# Predicting obesity reduction after implementing warning labels in Mexico: A modeling study

Ana Basto-Abreu[1], Rossana Torres-Alvarez[1], Francisco Reyes-Sánchez[1], Romina González-Morales[1], Francisco Canto-Osorio[1], M. Arantxa Colchero[2], Simón Barquera[3], Juan A. Rivera[4], Tonatiuh Barrientos-Gutierrez[1]*

1 Center for Population Health Research, National Institute of Public Health, Cuernavaca, Mexico, 2 Center for Health Systems Research, National Institute of Public Health, Cuernavaca, Mexico, 3 Center for Nutrition and Health Research, National Institute of Public Health, Cuernavaca, Mexico, 4 National Institute of Public Health, Cuernavaca, Mexico

* tbarrientos@insp.mx

**Data Availability Statement:** Baseline characteristics of the Mexican population are available at http://ensanut.insp.mx/. The final

## Abstract

### Background

In October 2019, Mexico approved a law to establish that nonalcoholic beverages and packaged foods that exceed a threshold for added calories, sugars, fats, trans fat, or sodium should have an "excess of" warning label. We aimed to estimate the expected reduction in the obesity prevalence and obesity costs in Mexico by introducing warning labels, over 5 years, among adults under 60 years of age.

### Methods and findings

Baseline intakes of beverages and snacks were obtained from the 2016 Mexican National Health and Nutrition Survey. The expected impact of labels on caloric intake was obtained from an experimental study, with a 10.5% caloric reduction for beverages and 3.0% caloric reduction for snacks. The caloric reduction was introduced into a dynamic model to estimate weight change. The model output was then used to estimate the expected changes in the prevalence of obesity and overweight. To predict obesity costs, we used the Health Ministry report of the impact of overweight and obesity in Mexico 1999–2023. We estimated a mean caloric reduction of 36.8 kcal/day/person (23.2 kcal/day from beverages and 13.6 kcal/day from snacks). Five years after implementation, this caloric reduction could reduce 1.68 kg and 4.98 percentage points (pp) in obesity (14.7%, with respect to baseline), which translates into a reduction of 1.3 million cases of obesity and a reduction of US$1.8 billion in direct and indirect costs. Our estimate is based on experimental evidence derived from warning labels as proposed in Canada, which include a single label and less restrictive limits to sugar, sodium, and saturated fats. Our estimates depend on various assumptions, such as the transportability of effect estimates from the experimental study to the Mexican population and that other factors that could influence weight and food and beverage consumption remain unchanged. Our results will need to be corroborated by future observational studies through the analysis of changes in sales, consumption, and body weight.

dataset after running our model is available at the open science framework, osf.io/5j4va (doi: 10.17605/OSF.IO/5J4VA). The mathematical model used to estimate the impact on body weight is freely available in the github within an R package named "bw" in https://github.com/INSP-RH/bw.

**Funding:** This project was funded by Bloomberg Philanthropies (https://www.bloomberg.org/; JAR received the grant). The funders had no role in study design, data collection and analysis, decision to publish, or preparation of the manuscript.

**Competing interests:** The authors have declared that no competing interests exist.

**Abbreviations:** BMI, body mass index; CHEERS, Consolidated Health Economic Evaluation Reporting Standards; CONAPO, Consejo Nacional de Población; INEGI, Instituto Nacional de Estadística y Geografía; MXP, Mexican peso; PAHO, Pan American Health Organization; pp, percentage point; SES, socioeconomic status; SFFQ, semiquantitative food frequency questionnaire; SSB, sugar-sweetened beverage.

## Conclusions

In this study, we estimated that warning labels may effectively reduce obesity and obesity-related costs. Mexico is following Chile, Peru, and Uruguay in implementing warning labels to processed foods, but other countries could benefit from this intervention.

## Author summary

### Why was this study done?

- In October 2019, the Mexican government approved new warning labels for packaged food and nonalcoholic beverages.

- Products that exceed a certain limit of calories, sugars, fats, trans fats, or sodium will now have a black octagon with an "excess of" label.

- Estimating the impact of these labels is necessary to translate this effort into expected reductions in intake and potential changes in body weight, obesity prevalence, and healthcare cost reductions.

### What did the researchers do and found?

- Adults in Mexico consume approximately 31% of their total energy intake from beverages and snacks. This study found that warning labels could reduce on average 36.8 kcal/person/day (23.2 kcal from beverages and 13.6 kcal from snacks).

- Using a mathematical model, we translated the expected caloric change into expected changes in body weight and obesity prevalence. Five years after the warning label implementation, obesity prevalence could be reduced by 14.7%, with respect to baseline, translating into 1.30 million cases of obesity reduced.

- Larger effects will be expected among males, young adults, and the middle and high socioeconomic status (SES) group.

- After five years, warning labels could save an estimated US$1.8 billion on obesity costs.

- Our model has some important limitations, such as using a Canadian estimate of the effect of warning labels. To capture a wider scope of effects, we used studies from Chile and Uruguay, which produced larger effects than the Canadian-based scenario.

### What do these findings mean?

- Warning labels have the potential to reduce the intake of nonessential high caloric food, reduce obesity, and lead to healthcare cost savings in Mexico.

- Warning labels could be considered by other countries with similar conditions, as part of their obesity control packages.

- These projections will need to be confirmed by future studies analyzing the change in food and beverage consumption and body weight after the implementation of the new warning labels.

## Introduction

High consumption of ultra-processed food and beverages is associated with increased caloric intake and weight gain [1,2]. Governments in many countries are establishing interventions to improve consumers' choices. In 2014, Mexico implemented a tax on sugar-sweetened beverages (SSBs) and nonessential highly caloric food, to discourage their consumption. The SSB tax was estimated to prevent 240 thousand cases of obesity and save US$4 on healthcare costs for every dollar spent in its implementation [3]. In October 2019, Mexico approved a new front of package labeling for nonalcoholic beverages and packaged food, under the law NOM-051 [4]. In January 2020, after a period of discussion with all interested stakeholders, the rules and specific terms of the law were approved by the regulatory committee, with minor changes [5]. The law includes all prepackaged food and beverages that add free sugars, fats (vegetable or animal), partially hydrogenated fats, or sodium (ingredient or additive) during the elaboration process. The law establishes a simple warning label with "excess of" calories, saturated fats, sodium, sugars, and trans fats (Fig 1). It will also include an additional legend in capital letters "contains sweeteners, not recommended in children" or "contains caffeine, avoid in children." The Mexican law based the additives limits on the recommendation of the Pan American Health Organization (PAHO) [4,6].

Warning labels have been proposed as a population-wide intervention to reduce the consumption of nonessential highly caloric food and beverages. Experimental studies have found that warning labels reduce the intention to purchase SSBs and snacks [7–12]. Using different designs of warning labels, the estimated reduction of calories ranged from 11.9% to 23.3% for beverages and from 5.5% to 11.7% for snacks [10–12]. A recent modeling study in the United States estimated that warning labels could reduce 25.3 kcal/day from SSBs, which was translated into a 3.1 percentage points (pp) reduction in the obesity prevalence [13]. These studies suggest that warning labels on food and beverages could be an effective intervention to improve the quality of diet and to reduce calories and obesity prevalence.

Considering that the consumption of processed food and beverages in Mexico is amongst the highest worldwide [2], we aimed to estimate the expected reduction in obesity prevalence

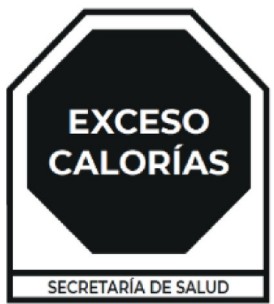 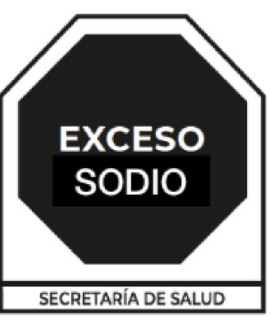 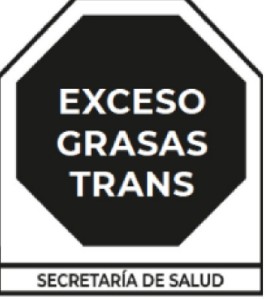 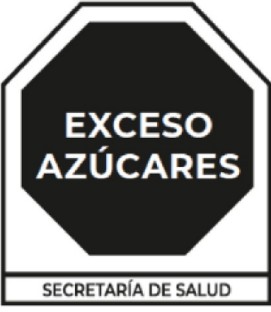 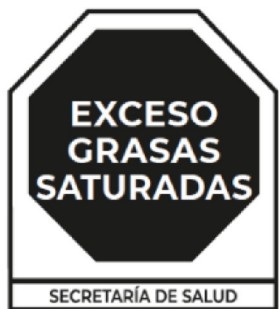

**Fig 1. Illustration of the warning labels proposed in Mexico under the NOM-051 [4].**

and obesity costs in Mexico by introducing warning labels, over 5 years among adults under 60 years of age.

## Methods

We used a simulation model to estimate the future impact on obesity and obesity costs that could be reduced by modifying the NOM-051 to introduce warning labels to packaged products in Mexico. We first estimated the impact on reduction of calories, body mass index (BMI), and obesity prevalence and then estimated the direct and indirect costs that could be reduced (Fig 2). Each step of the simulation strategy was detailed in the next subsections.

### Baseline intake of beverages and snacks and BMI

Baseline intake of beverages and snacks, and anthropometric data were collected using the 2016 National Health and Nutrition Survey (ENSANUT, from its Spanish acronym) with 6,511 individuals aged 20–59 years. The ENSANUT is a probabilistic multistage stratified cluster survey representative at national, regional, and rural/urban levels [15]. Within ENSANUT, we used the semiquantitative food frequency questionnaire (SFFQ) to estimate baseline intake of beverages and snacks. The SFFQ in 2016 was collected in all adults sampled for the ENSANUT up to 60 years old. It collects the consumption of 140 items over a week, including a variety of beverages and snacks that could have a warning label. Data include the number of days, times per day, serving size, and the number of times food was consumed during the 7 days prior to the interview. With the nutritional table, the quantity/day of beverages and snacks consumed was converted in daily energy intake for each subject (kcal/day/person), following standard procedures. The SFFQ was previously validated in Mexican adolescents and adults [17]. We selected beverages and snacks according to Acton and colleagues, 2019. The specific items selected in this study are included in Table A (S1 Appendix), and a detailed description is included in Section 2 (S1 Appendix).

The SFFQ was merged to anthropometric data with BMI, height, and weight. The resulting dataset presented 315 pregnant and lactating women, 137 individuals with missing data for weight or height, nine individuals with implausible values of BMI ($\geq$60 kg/m$^2$), one individual with an extreme value of SSBs intake (>16 servings/day [10]), and one individual with missing data on the primary sampling unit. All these individuals were excluded from the analysis, resulting in a final sample with 6,049 adults aged 20–59 years that, once survey weights were applied, represented 48,289,840 individuals in the adult population in Mexico.

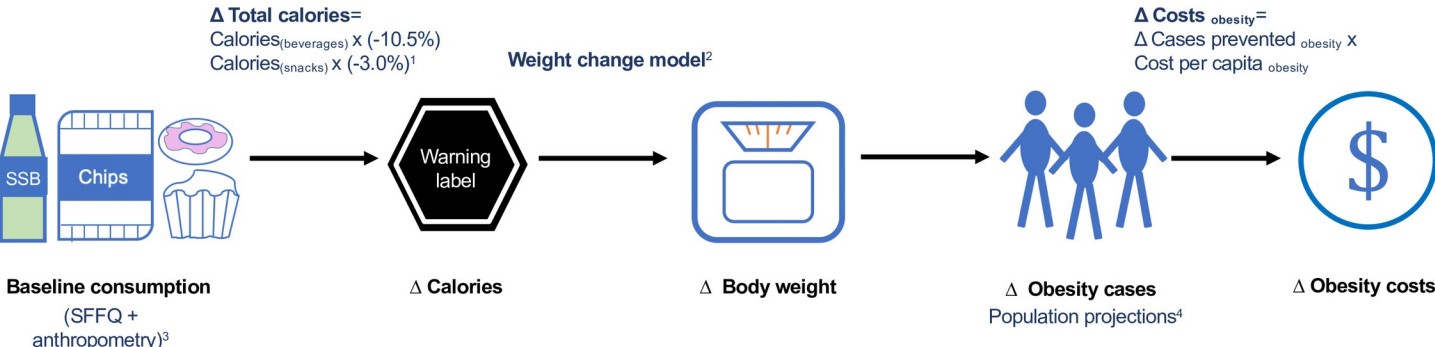

**Fig 2. Schematic illustration of the simulation strategy.** Sources of information: [1]Acton and colleagues, 2019; [2]Hall and colleagues, 2011; [3]ENSANUT 2016; [4]Population projections from CONAPO [10,14–16]. CONAPO, Consejo Nacional de Población; SFFQ, semiquantitative food frequency questionnaire; SSB, sugar-sweetened beverage.

### Reduction in energy intake from beverages and snacks

To estimate the expected impact of the warning labels on calories, we used an experimental marketplace study from Acton and colleagues using the "high in" warning label from Canada [10]. This study evaluated the effect of "high in" labels on purchases for beverages and snacks separately and estimated the reduction on energy and nutrients (sodium, saturated fats, and sugar), including individuals over 13 years and older [10]. We considered the reduction by 10.5% in beverages and by 3.0% in snacks, specifically for adults. The stratified results of Acton and colleagues by adults and adolescents are presented in Section 2.1 and Table E in S1 Appendix. This caloric change was observed between the experimental and control groups, and considers substitutions for other beverages or snacks. We considered a sodium reduction of 7.6% and 8.3% for beverages and snacks, respectively, for adults from Acton and colleagues as an input to the hall's model to estimate extracellular liquid for each individual [10].

In our study, we used the caloric effect, which includes the caloric reductions by specific labels, but other effects of saturated fats, sugars, and sodium are not considered. We assumed that the caloric effect using the Canadian warning labels would be similar to the Mexican warning labels, and that the reduction would occur at the beginning of the first year of implementation, remaining constant over time. Assumptions and source of information are described in Section 7 in S1 Appendix, and the parameters used are described in S1 Parameters.

### Reduction in body weight, BMI, and obesity prevalence

We estimated the expected impact on BMI attributable to warning label using a dynamic weight change model for adults, proposed by Hall and colleagues [14]. The model estimates the change in body weight of each individual at time *t*, considering changes in extracellular fluid, glycogen, and fat and lean tissues triggered by the change in consumption of food and beverages with the warning labels, maintaining the physical activity constant. We ran the model for each year and up to five years to obtain the complete effect of caloric change in weight changes. In Mexico, this model was previously used in adults to estimate the expected impact of SSB tax and reformulation to reduce sugar content [17,18]. The model structure is described in S1 Parameters and a full description of the model can be found elsewhere [14].

### Reduction in cases with obesity

After estimating changes in obesity prevalence, we translated these changes in cases averted over 5 years. For that purpose, we used population projections from the Consejo Nacional de Población (CONAPO) (in English, National Population Council), for adults 20–59 years old, from 2019 to 2023 [16]. To estimate total cases of obesity over the simulated period, we multiplied the obesity prevalence estimated using ENSANUT 2018 for adults aged 20–59 (37.8%) and assumed a steady state for the obesity prevalence (i.e., prevalence does not change for any other reason but the warning label intervention). By multiplying the prevalence change by the estimated number of adults with obesity in Mexico, we estimated the cases of obesity that could be averted in the period 2019–2023.

**Sensitivity analysis.** We conducted a sensitivity analysis to estimate the uncertainty range around the impact of warning labels in obesity cases. For beverages, we estimated three potential alternative scenarios. The first is based on observed reductions in an observational study in Chile; it assumes a 7.5% reduction in caloric intake from beverages after the implementation of "high in" labels and considers substitution for not "high in" beverages [19]. The second and third scenarios are based on a stratified estimation from the same observational study in Chile, having a 27.5% decrease in "high in" and a 10.8% increase in "not high in" beverages [19]. The

second scenario categorizes "high in" or "not high in" beverages following the thresholds defined by the Chilean law, while the third scenario follows the thresholds proposed in the Mexican law. The limits thresholds proposed by Chile and Mexico are included in Table F and Table I in S1 Appendix. For snacks, we used an experimental study from Uruguay, with 199 subjects from a university in Montevideo, which estimated an 11.68% caloric reduction and a 50.17% sodium reduction after the implementation of warning labels [11]. More information on how beverages and snacks were selected can be found in Tables A and B in S1 Appendix. More information on model parameters are included in S1 Parameters.

### Reduction in healthcare costs

Obesity costs were obtained over 5 years after implementing the warning labels. For that, we used the financial report of the impact of overweight and obesity in Mexico 1999–2023 generated by the Health Ministry [20]. Direct costs use the health system perspective, in which out-of-pocket expenses are not included. Indirect costs use the individual perspective, including premature death, absenteeism, and caregiver expenses from the individual perspective and pension allowance for work incapacity and disability from a social security perspective. As obesity costs in the report were obtained for the total population, we limited the obesity costs to the target population (20–59 years). More information about the source of data and a detailed description of cost estimations is included in Section 5 in S1 Appendix.

We projected the estimated obesity costs in 2014 to the year 2019 using the price deflator based on the National Consumer Price Index of the Instituto Nacional de Estadística y Geografía (INEGI) (in English, the National Institute of Statistics and Geography) [21] and discounted 3% yearly to estimate the costs over the next 4 years, following standard procedures in a cost-effectiveness analysis [22]. Finally, we multiplied the absolute reduction in obesity prevalence by the obesity costs during the 5 years to estimate the total direct (healthcare) costs and indirect costs (premature deaths, work absenteeism, and others) that could be reduced among adults 20 to 59 years old. Costs in Mexican pesos (MXP) were converted to American dollars using the 2019 average exchange rate (1 MXP = 0.05190017 US$) [23]. A detailed description of cost estimations is included in Section 5 in S1 Appendix. This study is reported as per the Consolidated Health Economic Evaluation Reporting Standards (CHEERS) guideline (S1 Checklist).

### Results

Table 1 presents the baseline intake of beverages and snacks among Mexican adults in 2016. We estimated a baseline intake of 220.9 kcal/day from beverages and 453.1 kcal/day from snacks, which together account for 31.1% of the total energy intake. The expected caloric change after the warning labeling was estimated to be −36.8 kcal/day (−23.2 kcal/day from beverages and −13.6 kcal/day from snacks).

Table 2 presents the impact of warning labels in Mexico. Body weight is expected to decrease by 1.68 kg (1.05 kg from beverages and 0.63 kg from snacks) and BMI by 0.65 kg/m$^2$. This reduction would imply a 4.98 pp reduction in obesity over five years, representing 1.3 million cases of obesity among adults under 60 years.

**Table 1. Baseline intake of beverages and snacks and its expected change in calories and sodium attributable to warning labels among adults 20–59 years old.**

| Food category | Baseline energy intake (kcal/day/person, CI 95%) | Baseline intake as a percentage of total energy intake (%) | Expected change in energy intake after the labeling (kcal/day/person, CI 95%) |
|---|---|---|---|
| Beverages | 220.9 (208.6 to 233.2) | 9.9 | −23.2 (−24.5 to −21.9) |
| Snacks | 453.1 (435.5 to 470.7) | 21.2 | −13.6 (−14.1 to −13.1) |
| Total | 674.0 (650.4 to 697.6) | 31.1 | −36.8 (−38.3 to −35.3) |

**Table 2. Weight, BMI, obesity, and prevalence change attributable to warning labels over 5 years among adults 20–59 years old.**

| Health outputs | Baseline, CI 95% | Change over 5 years for beverages, CI 95% | Change over 5 years for snacks, CI 95% | Total change over 5 years, CI 95% |
|---|---|---|---|---|
| Body weight (kg) | 72.38 (71.60 to 73.16) | −1.05 (−1.11 to −1.00) | −0.63 (−0.65 to −0.60) | −1.68 (−1.75 to −1.61) |
| BMI (kg/m$^2$) | 28.34 (28.08 to 28.60) | −0.41 (−0.43 to −0.38) | −0.25 (−0.26 to −0.24) | −0.65 (−0.68 to −0.63) |
| Obesity prevalence (pp) | 33.81 (31.54 to 36.07) | −2.92 (−3.67 to −2.16) | −1.83 (−2.45 to −1.22) | −4.98 (−6.03 to −3.93) |
| Cases of obesity (thousand people) | 26,174 | 764 | 479 | 1,303 |

Abbreviations: BMI, body mass index; pp, percentage point

Fig 3 presents the impact of warning labels in Mexico in obesity prevalence over 5 years by age groups, sex, and socioeconomic status (SES) groups. Expected reductions in obesity, with respect to baseline, were larger among adults under 40 years (−17.2%) in comparison with individuals between 40–59 years (−12.2%). Also, males presented larger obesity reductions (−16.2%) in comparison to females (−13.8%). Middle and high SES (−15.1% and 15.5%) presented larger obesity reductions in comparison with low SES (−12.6%). The specific absolute and relative yearly changes can be found in Table T in S1 Appendix.

Fig 4 presents the range of potential reduction in obesity cases varying the parameter effect of warning labels on beverages and snacks. We found that warning labels could reduce from

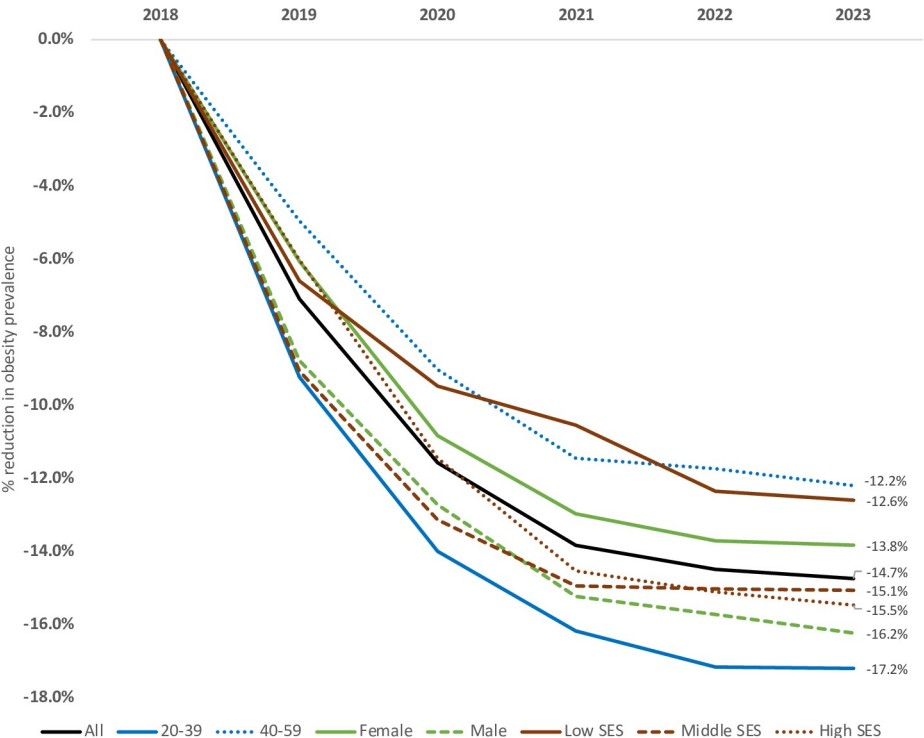

**Fig 3. Percent change (%) of obesity prevalence over 5 years by age group, sex, and SES. SES, socioeconomic status.**

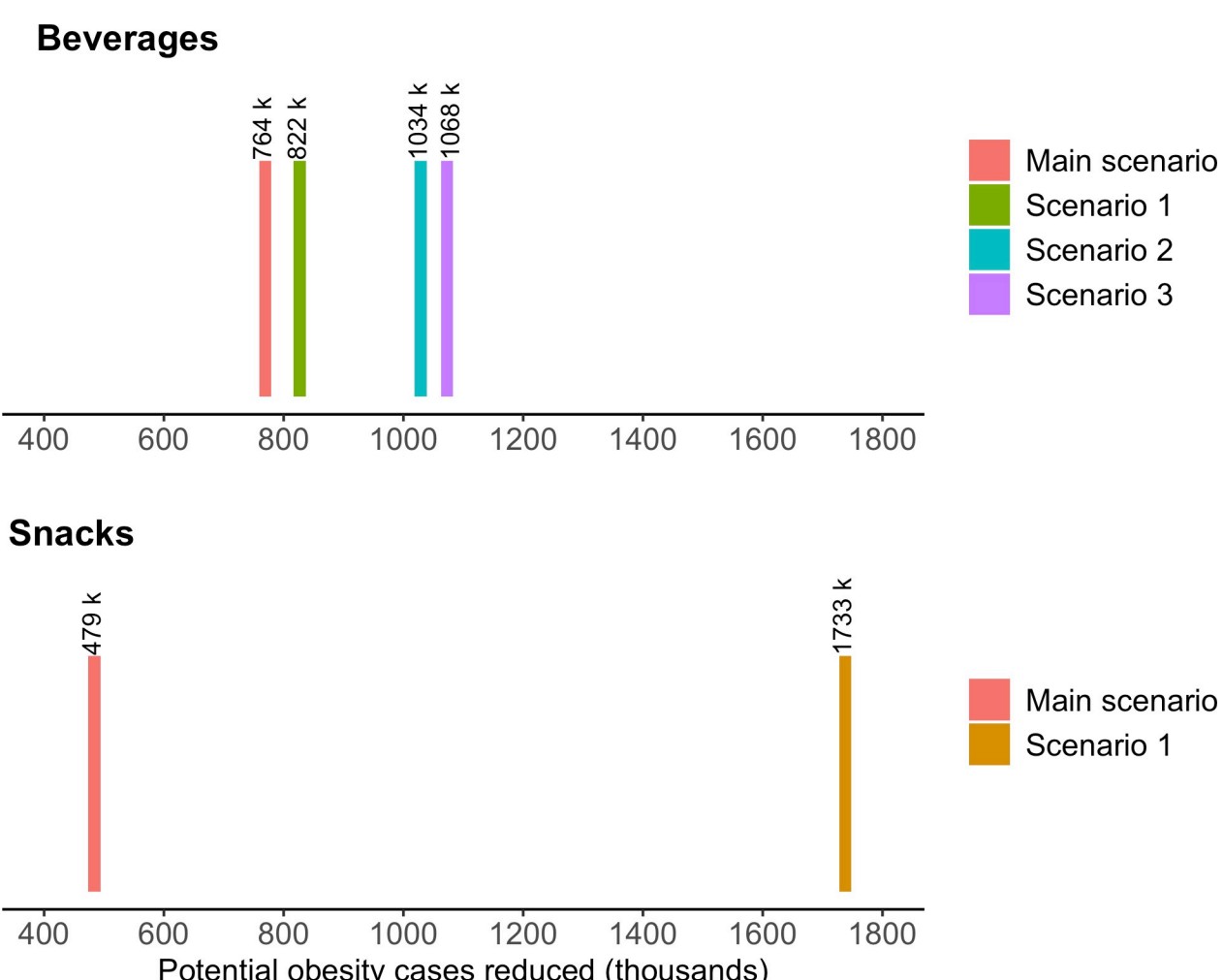

**Fig 4. Potential impact on obesity cases reduced using different scenarios, stratified by beverages and snacks.** Main scenario: Effect estimate (10.5% and 3.0% caloric reduction for beverages and snacks, respectively) based on a Canadian experimental study. Scenario 1: Effect estimate (7.5% caloric reduction) based on Chilean observational study. Scenario 2: Effect estimate (caloric reduction by 27.5% in "high in" beverages and caloric increase by 10.8% in not "high in" beverages) based on Chilean observational study and adding the Chilean limits for the first stage of the law (Table F in S1 Appendix). Scenario 3: Effect estimate (caloric reduction by 27.5% in "high in" beverages and caloric increase by 10.8% in not "high in" beverages) based on Chilean observational study and adding the Mexican limits (Table H in S1 Appendix). Scenario 1 snacks: Effect estimate (11.68% caloric reduction for snacks) based on Uruguayan study.

764 thousand (main scenario) to 1.07 million of people with obesity over 5 years through reducing beverages intake. In addition, warning labels could reduce from 479 thousand (main scenario) to 1.73 million people with obesity through reducing snacks intake.

Table 3 presents the direct and indirect costs that could be reduced over 5 years after implementing the warning labels in Mexico. Obesity costs for the adult population under 60 years of age was estimated to be US$44.5 billion, from which 40% were indirect costs. With the warning labels, we estimated that 5 years after implementation, we would be able to potentially save US$1.8 billion, including US$742 million from indirect costs.

## Discussion

We aimed to estimate the potential impact of warning labels in beverages and snacks in Mexico over the obesity prevalence and obesity-related costs. The warning labels were

**Table 3. Obesity costs prevented due to warning labels in Mexico after 5 years among adults 20–59 years old.**

| Cost outputs over 5 years | Baseline costs | Change in costs |
|---|---|---|
| Direct costs of obesity (million US$) | 26,591 | −1,100 |
| Indirect costs of obesity (million US$) | 17,944 | −742 |
| Total costs reduced (million US$) | 44,535 | −1,842 |

All costs are from 2019.

estimated to reduce 36.8 kcal/day, which could reduce 4.98 pp in the obesity prevalence among adults under 60 years (−14.7%, with respect to baseline). Converted to absolute numbers over 5 years, this strategy was estimated to reduce 1.3 million cases of obesity and save US$1.8 billion in obesity costs.

Some countries have already implemented warning labels for foods and beverages, including Chile, Peru, and Uruguay [24]. To our knowledge, none of these countries have estimated yet the observed impact of warning labels on obesity prevalence. A modeling study in 2019 estimated the future impact of warning labels in the US, considering only the effect on beverages [13]. The warning labels modeled consisted of a rectangle with the text "WARNING: Beverages with added sugar contribute to tooth decay, diabetes, and obesity." To inform the expected changes in calories, the authors used as the main scenario the effect of warnings on willingness to pay or on purchasing beverages among adults (−12.7%), while we used an experimental study in Canada with "high in" labels (−10.5% for beverages). As a result, the estimated change in calories for beverages was higher in the US than in Mexico (25.3 kcal/day versus 23.2 kcal/day). Using a sample distribution for caloric compensation, the US study found a total caloric change of 31.2 kcal/day, while we included the caloric change from snacks, predicting a total caloric change of 36.8 kcal/day. The caloric change was translated into identical expected reductions in BMI but into larger reductions in obesity prevalence (3.1 pp versus 4.98 pp in our study). Differences could be due to the BMI distribution, with more people in Mexico being near the obesity cutoff. Another modeling study estimated the impact of traffic-light nutrition labels on solids among adult Australians [25]. The authors estimated a caloric change of −36.7 kcal/day for males and −21.1 kcal/day for females, in comparison with −13.6 kcal/day observed in snacks in our study. Differences are explained by the expected caloric change after the implementation of warning labels for solids: −10% in the Australian study, based on three studies, compared to −3.0% used in our study [26–28]. Overall, we consider that our assumptions are conservative and based on the best available experimental evidence for warning labels.

To our knowledge, our study is the first to estimate the obesity costs that could be reduced by implementing the warning labels. We estimated that warning labels could save more than US$1.8 billion in obesity costs, from which $1.1 billion were direct and $742 million were indirect costs. The "traffic-light" nutrition labeling was estimated to save around US$367 million in obesity prevention [25]. Our direct cost savings are nearly triple because we are also considering reductions in beverages. Other differences may be due to the following: (1) costs in the Australian study are based in 2003, whereas in our study, in 2019, (2) the Mexican adult population is over 5 times the Australian population, and (3) obesity prevalence is higher in Mexico than in Australia. While estimating the cost-effectiveness of the warning labels is beyond the scope of this article, considering that the cost estimated by the industry to implement the nutrition labeling Daily Nutrition Guidelines in Mexico was US$312.6 million (cost translated to dollars in 2019) [29], warning labels would save US$5.9 for each dollar spent in implementing, being a cost-benefit intervention.

Our modelling study presents several limitations and several layers of uncertainty. To inform the caloric change after the warning label, we used an experimental study that employed a single red warning label and nutrition thresholds proposed by Health Canada. While we used the baseline caloric intake from Mexico, the effect size from Canada (−10.5% for beverages and −3% for snacks) could change in the Mexican population. The Mexican law has more restrictive limits than the proposed law in Canada (Section 2 in S1 Appendix), which could result in more products having the warning label and a higher caloric reduction. As transporting the effect from Canada to Mexico is challenging, we varied the caloric reduction parameter using estimates from Chile (for beverages) and from Uruguay (for snacks). For beverages, we estimate an obesity case reduction ranging from 764,000 to 1,068,000, and for snacks ranging from 479,000 to 1,733,000 of obesity cases. We decided to keep the Canadian scenario as our main analysis because it is experimental and was the only source with effect size for both beverages and snacks. Also, the Canadian scenario produced the most conservative results. Our model relies on a steady-state assumption. Because it is an individual-level simulation model and a closed cohort, we could not integrate into our estimation process the increasing obesity trend that the Mexican adult population is still experiencing (in 2016, the obesity prevalence was 33.3% compared to 36.1% in 2018) [30]. Lack of consideration for the increasing obesity trend would lead to an underestimation of the total number of obesity cases reduced, because the estimated reduction over time would be multiplied by fewer obesity cases.

Our model focuses on the estimation of the expected body weight change as result of a reduction in caloric intake associated with warning labels. However, warning labels could produce health benefits that we are not modeling. For instance, the warning label is expected to reduce 8.3% of sodium consumption from snacks, which should have a major impact on reducing blood pressure, cardiovascular diseases, hypertension, and overall mortality. Also, expected reductions in saturated fat or sugars (not considered in this analysis) would have an impact on diabetes or atherosclerosis that are not mediated through obesity; thus, other health benefits need to be independently assessed and considered. Our analysis is limited to the direct effect of warning labels on purchase intention and does not take into consideration the potential reformulation that could be induced by the threshold at which a label is implemented. Reformulation is a voluntary strategy that the industry has been employing to reduce sodium, saturated fats, and added sugars to avoid high taxes or warning labels. While Chile observed very small reductions in critical nutrients before the law came into effect (June 2016), to our knowledge, Chile has not yet evaluated the reformulation effect after implementing the warning labels [31]. However, 29% and a 40% reductions in sugar content were observed on taxed beverages in the United Kingdom and South Africa, respectively [32,33].

Other limitations regarding the nutrition assessment tool used (SFFQ from 2016) need to be mentioned. The SFFQ 2016 was collected from May to August 2016, during the summer months in Mexico, when beverage consumption increases in comparison to winter, which could lead to overestimate the impact of the warning labels, compared to the annual SSB consumption. However, self-reported nutrition tools such as the SFFQ tend to underestimate the consumption of highly caloric food and beverages, and the underestimation is usually higher among people with higher BMI. If consumption is underestimated, our predictions would be conservative. The SFFQ over 7 days in comparison with 24-hour dietary recall may have larger measurement error due to recall bias. However, we decided to use the SFFQ in order to capture the usual intake by an individual and because the 24-hour dietary recall from ENSANUT 2016 had a smaller sample size (four times less). Finally, we are assuming that producers and supermarkets will comply 100% with the law. A previous study in Chile observed a high level of compliance from producers, with 32 out of 41 products implementing the warning labels; however, full compliance was not attained [34].

Our study models the potential impact of warning labels, relaying in a large set of assumptions that could diverge from the real-life conditions of implementation. Thus, it will be important to conduct future studies to contrast our projections against observational data. Building upon the literature from SSBs, we propose for studies to analyze different end points, such as changes in sales, purchases, and consumption of labeled versus unlabeled products [35,36]. It will also be important to analyze changes in body weight, incidence of chronic diseases, and biomarkers, although these studies will be difficult to conduct given that labels will be implemented at the same time in the whole nation. The uncertainty about the lag between the intervention and the expected effects, and the magnitude of the expected change (−1.68 kg, according to our estimates) could be difficult to detect even in large, well-conducted longitudinal studies.

The obesity epidemic is one of the biggest public health challenges for modern societies. A single intervention such as warning labels will not be the silver bullet to eradicate obesity. Structural interventions are needed to counter the increasing trends in body weight that have been observed in Mexico and other countries. Mexico has been at the forefront of implementation of structural interventions to curb the obesity epidemic, such as the tax to SSBs and nonessential high caloric food; the SSB tax was estimated to reduce on average 0.15 kg/m$^2$ in the adult population [18]. Another modeling study estimated that SSB tax could reduce 0.21 pp in obesity prevalence, while the warning label is expected to reduce 4.9 pp in adults below 60 years. Warning labels seem capable of reducing body weight through an independent mechanism, being likely additive to the impact of taxes; future analyses are needed to understand how taxes and warning labels interact to produce changes in consumption, so that any potential overlap or synergy can be considered when designing these interventions.

Warning labels on processed food and beverages are expected to effectively reduce obesity (−14.7%, with respect to baseline) and could save US$1.8 billion in obesity costs. Latin American countries are following Chile, implementing warning labels to inform consumers, reduce market failures, and facilitate healthier dietary choices. While more efforts will be needed to curb the obesity epidemic in Mexico, it is clear that warning labels have the potential to benefit the Mexican population by preventing an important number of obesity cases and costs.

## Supporting information

**S1 Checklist. Items to include when reporting economic evaluations of health interventions, from CHEERS.** CHEERS, Consolidated Health Economic Evaluation Reporting Standards
(PDF)

**S1 Appendix. Additional information on the baseline consumption of beverages and snacks, weight change model, obesity cases, obesity costs, complementary results, and assumptions.**
(PDF)

**S1 Parameters. Model structure and model parameters, with their mean values and range when available, sources, and caveats.**
(PDF)

## Acknowledgments

We are grateful to Rachel Acton, from the School of Public Health and Health Systems, University of Waterloo, for her kindness and valuable feedback. We would like to express our gratitude for allowing us to share her results, stratified by adults and adolescents, in our S1

Appendix. We are also thankful to Ana Munguia and Lizbeth Tolentino-Mayo from the National Institute of Public Health, Mexico, for their valuable feedback on the NOM-051 that was approved in Mexico.

## Author Contributions

**Conceptualization:** Ana Basto-Abreu, Rossana Torres-Alvarez, Francisco Canto-Osorio, M. Arantxa Colchero, Simón Barquera, Juan A. Rivera, Tonatiuh Barrientos-Gutierrez.

**Data curation:** Rossana Torres-Alvarez, Francisco Reyes-Sánchez.

**Formal analysis:** Ana Basto-Abreu, Rossana Torres-Alvarez, Francisco Reyes-Sánchez.

**Funding acquisition:** Simón Barquera, Tonatiuh Barrientos-Gutierrez.

**Investigation:** Ana Basto-Abreu, Rossana Torres-Alvarez, Francisco Reyes-Sánchez, Romina González-Morales, Francisco Canto-Osorio, M. Arantxa Colchero, Simón Barquera, Juan A. Rivera, Tonatiuh Barrientos-Gutierrez.

**Methodology:** Ana Basto-Abreu, Rossana Torres-Alvarez, Francisco Reyes-Sánchez, Romina González-Morales, Francisco Canto-Osorio, M. Arantxa Colchero, Tonatiuh Barrientos-Gutierrez.

**Project administration:** Juan A. Rivera, Tonatiuh Barrientos-Gutierrez.

**Resources:** Juan A. Rivera.

**Supervision:** M. Arantxa Colchero, Simón Barquera, Juan A. Rivera, Tonatiuh Barrientos-Gutierrez.

**Validation:** Ana Basto-Abreu, Tonatiuh Barrientos-Gutierrez.

**Visualization:** Ana Basto-Abreu, Romina González-Morales, Francisco Canto-Osorio, Simón Barquera, Tonatiuh Barrientos-Gutierrez.

**Writing – original draft:** Ana Basto-Abreu, Rossana Torres-Alvarez, Francisco Reyes-Sánchez, Romina González-Morales, Tonatiuh Barrientos-Gutierrez.

**Writing – review & editing:** Ana Basto-Abreu, Rossana Torres-Alvarez, Francisco Reyes-Sánchez, Romina González-Morales, Francisco Canto-Osorio, M. Arantxa Colchero, Simón Barquera, Juan A. Rivera, Tonatiuh Barrientos-Gutierrez.

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
