## [Editor Report · Decision Letter 0]

20 Feb 2020

Dear Dr Barrientos-Gutierrez, 

Thank you for submitting your manuscript entitled "Expected obesity reduction after implementing warning labels in Mexico: a modeling study" for consideration by PLOS Medicine.

Your manuscript has now been evaluated by the PLOS Medicine editorial staff [as well as by an academic editor with relevant expertise] and I am writing to let you know that we would like to send your submission out for external peer review.

Kind regards,

Adya Misra, PhD,

Senior Editor

PLOS Medicine

---

## [Decision Letter · Decision Letter 1]

18 May 2020

Dear Dr. Barrientos-Gutierrez,

Thank you very much for submitting your manuscript "Expected obesity reduction after implementing warning labels in Mexico: a modeling study" (PMEDICINE-D-20-00367R1) for consideration at PLOS Medicine. 

[LINK]

In light of these reviews, I am afraid that we will not be able to accept the manuscript for publication in the journal in its current form, but we would like to consider a revised version that addresses the reviewers' and editors' comments. Obviously we cannot make any decision about publication until we have seen the revised manuscript and your response, and we plan to seek re-review by one or more of the reviewers. 

We expect to receive your revised manuscript by May 31 2020 11:59PM. Please email us (plosmedicine@plos.org) if you have any questions or concerns.

We look forward to receiving your revised manuscript. 

Sincerely,

Adya Misra, PhD

Senior Editor 

PLOS Medicine

plosmedicine.org

Abstract

Background-please provide further info about the law on warning labels-whether it is single or multi labels etc and which products require these warning labels

Methods and findings section needs more information about the model, assumptions/data used to create the model

Please clarify what data sources were used to estimate costs

Last sentence of the methods and findings section should include a limitation of the study design/methodology

Abstract Conclusions:

* Please address the study implications without overreaching what can be concluded from the data; the phrase "In this study, we observed ..." may be useful.

* Please interpret the study based on the results presented in the abstract, emphasizing what is new without overstating your conclusions.

* Please avoid vague statements such as "these results have major implications for policy/clinical care". Mention only specific implications substantiated by the results.

* Please avoid assertions of primacy ("We report for the first time....")

Author summary

Please add a full stop after the references in square brackets

Introduction

Please can you provide brief details of the law on warning labels (as stated above). Please add details of which food/drinks are eligible 

Please clarify in text if the law NOM-051 is the SSB taxation law that was implemented in 2014

Please provide a reference for “Considering that the consumption of industrialized food products in Mexico is amongst the highest worldwide”

Please rephrase “obesity costs that could be averted in the country by introducing warning labels, over 5 years among adults under 60 years of age” to more accurately reflect the aim of your work. It is also not clear whether obesity costs can be fully averted with warning labels, so I would avoid this type of language

Methods

Please include the citation for the study Acton et al when it is first mentioned on page 4

You may wish to comment why you used estimates from Canada- since the population is quite different as is the warning label

“Acton, et al., observed that adults experienced smaller caloric reductions (10.5% in beverages

and 3.0% in snacks) compared to adolescents (16.7% in beverages and 15.9 % in snacks) (data provided by the authors)” – if this information is not published, please include this data as SI files or remove this information. Please note the methods section should only include the methodology of your study, so the middle paragraph on page 5 should be removed

Section 1.3 should be moved up in the methods section as it provides important information about the model used

Please state the Spanish name of the national population council and National Institute of Statistics and Geography, providing the English names in brackets along with acronyms for clarity 

Please reconsider the repeated use of the phrase “costs averted” or “averted”. There can be a reduction in cost but I am not sure these can be completely averted by the use of warning labels. 

Please format the bibliography in Vancouver style

Please present and organize the Discussion as follows: a short, clear summary of the article's findings; what the study adds to existing research and where and why the results may differ from previous research; strengths and limitations of the study; implications and next steps for research, clinical practice, and/or public policy; one-paragraph conclusion.

Please ensure that the study is reported according to the STROBE/CHEERS guideline, and include the completed checklists as Supporting Information. When completing the checklist, please use section and paragraph numbers, rather than page numbers. Please add the following statement, or similar, to the Methods: "This study is reported as per the XXX guideline (S1 Checklist)."

Comments from the reviewers:

Reviewer #1: In this study, the authors attempt to estimate the expected change in the obesity prevalence and obesity costs that could be averted in Mexico by introducing warning labels, over 5 years among adults under 60 years of age.

Under Abstract (and throughout the manuscript):

"Our estimate is based on experimental evidence derived from warning labels as proposed in Canada, which include a single label and less restrictive limits to sugar, sodium and saturated fats; however, the Mexican warning label law is much stricter and includes up to five warning labels per product, thus, our estimates are conservative."

The different labeling protocol does not necessarily make this analysis more conservative. There could be many factors at play which may have an impact on effectiveness of labeling. 

"Mexico is following Chile, Peru and Uruguay in implementing warning labels to processed foods, but other countries could benefit from this intervention."

Is it possible to use evidence gathered from Chile, Peru and Uruguay to estimate the effectiveness and impact of labeling in Mexico?

Under Methods:

"All these individuals were excluded from the analysis, resulting in a final sample with 6,050 adults aged 20-59 years, representing 48,290,327 individuals. "

Can the authors please clarify this sentence?

"We assumed that the caloric effect using the Canadian warning labels would be similar to the Mexican warning labels, and that the reduction would occur at the beginning of the first year of implementation, remaining constant over time."

Is Canada similar to Mexico in terms of cultural attitudes towards snacking, as well as quantity and type of snacks consumed?

"...assumed steady state for the obesity prevalence (i.e., prevalence does not change for any other reason, but the warning label intervention)."

Does this account for longitudinal trends of obesity seen in Mexico (i.e. is there an otherwise increasing prevalence in obesity)?

Under Results:

The tables are clear and easy to interpret. 

Table 3 would benefit from being displayed as a graph or chart, with lines or columns representing changes over time for age group, sex and socioeconomic status groups.

The analysis and results would be far more robust and could be interpreted with more appropriate uncertainty if variability in assumptions and sensitivity analyses were accounted for in the simulated estimates. 

Furthermore, the results of this additional assessment of uncertainty could then be included in Table 4.

Under Supplementary Information:

The S1 Appendix is a useful resource for the reader to better understand the model and underlying assumptions.

Reviewer #2: Expected obesity reduction after implementing warning labels in Mexico: a modeling study

1. This article models the potential impact of warning labels for beverages and snacks on obesity and obesity-related costs in Mexico. The topic is important, and the motivation is enhanced by the fact that such a policy recently has passed. The study is well-written and the methods and results are quite clear.

2. The article does not use a checklist for completeness. If relevant, formally or informally, it would be good to reference a checklist such as CHEERS or an authoritative source for methods, such as the second panel report for cost-effectiveness analysis (Neumann et al.). Even though this is not a formal cost-effectiveness analysis, that report may be useful. It is referenced in this article just as authority for 3% discount rate, but the application may be more widespread.

3. For example, current practice for such articles may require use of probabilistic or deterministic sensitivity analysis, which the article does not have. The article argues essentially that it has placed a bound on the plausible estimates, so any sensitivity to assumptions is in only one direction, but this cannot be determined with confidence. In the discussion, greater uncertainty should be acknowledged. In such articles, the output depends on uncertain in several layers of inputs.

4. In particular, the article is especially highly reliant on the ability to apply an estimate from a labeling experiment in Canada (Acton) to nationwide impact in Mexico. The Acton study is sufficiently central that more features of methods and limitations should be briefly summarized in this article. Uncertainty bounds in the Acton estimates could be considered in sensitivity analysis in this article. It is not clear if calorie changes from that study can be maintained without modification nationwide for intake in Mexico. It should be noted that the Acton study appears to be largely a hypothetical experiment, with a randomly assigned purchase offering just partial realism for actual economic impact for respondents. If the Acton study uses only purchase as the outcome, rather than daily intake, then there may be compensation at other times of day, leading daily intake proportional effects to be smaller.

5. Because the policy is actually being implemented, the article could discuss how future evidence from the field can be compared to these modeling estimates. In a scientific spirit, one could point out whether and how the estimates in this study can and should be corroborated.

[LINK]

---

## [Editor Report · Decision Letter 2]

4 Jun 2020

Dear Dr. Barrientos-Gutierrez,

Thank you very much for re-submitting your manuscript "Expected obesity reduction after implementing warning labels in Mexico: a modeling study" (PMEDICINE-D-20-00367R2) for review by PLOS Medicine.

I have discussed the paper with my colleagues and the academic editor and it was also seen again by reviewers. I am pleased to say that provided the remaining editorial and production issues are dealt with we are planning to accept the paper for publication in the journal.

[LINK]

We look forward to receiving the revised manuscript by Jun 11 2020 11:59PM. 

Sincerely,

Adya Misra, PhD

Senior Editor 

PLOS Medicine

plosmedicine.org

Requests from Editors:

Title- Please replace "expected" with "predicting" given it’s a modelling study. 

Please add a space in front of the square bracket for references throughout the submission

Line 118 ‘industrialised’ – I wonder if factory made or processed might be better

CHEERS checklist- please remove page and line numbers as these are likely to change. Please use paragraphs and sections instead

Abstract

Please define “pp” on first view 

The limitations have not ben clearly laid out. It is not sufficient to say that the study findings need to be confirmed but please include limitations of your study design. For instance- the use of evidence from Canada applied to Mexico might have some limitations. Please do add these to the author summary as well

Line 95- I’m not sure about market failures. Could you please clarify and revise as needed

Line 352- I think there have been a few studies describing reformulation due to SSBs. I am not asking you to cite all the evidence here necessarily but it would be incorrect to say no studies have been published. Please let me know if I have misinterpreted this sentence

Did your study have a prospective protocol or analysis plan? Please state this (either way) early in the Methods section.

Please provide a complete list of model parameters, including clear and precise descriptions of [the meaning of each parameter, together with the values or ranges for each, with justification or the primary source cited, and important caveats about the use of these values noted]. Please discuss the scientific rationale for this choice of model structure and identify points where this choice could influence conclusions drawn. Please also describe the strength of the scientific basis underlying the key model assumptions.

Throughout, please clarify if the costs calculated are from the perspective of the health system or the individual.

Comments from Reviewers:

[LINK]

---

## [Editor Report · Decision Letter 3]

24 Jun 2020

Dear Dr. Barrientos-Gutierrez, 

On behalf of my colleagues and the academic editor, Dr. Karine Clément, I am delighted to inform you that your manuscript entitled "Predicting obesity reduction after implementing warning labels in Mexico: a modeling study" (PMEDICINE-D-20-00367R3) has been accepted for publication in PLOS Medicine. 

PRODUCTION PROCESS

PRESS

PROFILE INFORMATION

Thank you again for submitting the manuscript to PLOS Medicine. We look forward to publishing it. 

Best wishes, 

Adya Misra, PhD

Senior Editor 

PLOS Medicine

plosmedicine.org